# Eyes on incivility in surgical teams: Teamwork, well-being, and an intervention

**Cheri Ostroff**[1]*, **Chelsea Benincasa**[2], **Belinda Rae**[1], **Douglas Fahlbusch**[3], **Nicholas Wallwork**[4]

**1** University of South Australia Centre for Workplace Excellence, Adelaide, Australia, **2** University of South Australia Rosemary Bryant AO Research Centre, Adelaide, Australia, **3** University of South Australia Clinical and Health Sciences, Adelaide, Australia, **4** Sportsmed, South Australian Hospital, Adelaide, Australia

* cheri.ostroff@unisa.edu.au

**Data Availability Statement:** Data are available on OSF: https://doi.org/10.17605/OSF.IO/67HNV.

**Funding:** This project was supported by the Australian Research Council Grant, Project DP160101313 (CO received the award). The

## Abstract

Incivility in surgery is prevalent and negatively impacts effectiveness and staff well-being. The purpose of this study was to a) examine relationships between incivility, team dynamics, and well-being outcomes, and b) explore a low-cost intervention of 'eye' signage in operating theater areas to reduce incivility in surgical teams. A mixed methods design was used in an orthopedic hospital. Surveys of incivility, teamwork, and well-being were administered three months apart in a small private hospital. An intervention of signage with eyes was placed in the theater area after administration of the first survey, using a pretest-posttest design. Participants also responded to an open-ended question about suggestions for improvements at the end of the survey which was then thematically analyzed. At the individual level ($n = 74$), incivility was statistically significantly related to team dynamics which in turn was significantly related to burnout, stress, and job attitudes. At the aggregate level, reported incivility was statistically significantly lower after the 'eye' sign intervention. Thematic analysis identified core issues of management behaviors, employee appreciation, communication, and work practices. Incivility in surgical teams has significant detrimental associations with burnout, stress, and job attitudes, which occurs through its impact on decreased team dynamics and communication. A simple intervention that evokes perceptions of being observed, such as signage of eyes in theater areas, has the potential to decrease incivility at least in the short term, demonstrating that incivility is amenable to being modified. Additional research on targeted interventions to address incivility are needed to improve teamwork and staff well-being.

## Introduction

Interpersonal communication characterized by incivility (e.g., rudeness, derision, insulting remarks, humiliation, ignoring someone) can have detrimental consequences on employee effectiveness in a wide variety of jobs, organizations, and industries [1]. These small acts of rudeness that violate basic standards of respect can have large impacts on well-being, particularly when they accumulate over time [2]. Incivility in healthcare is prevalent with the majority

funders had no role in study design, data collection and analysis, decision to publish, or preparation of the manuscript.

**Competing interests:** The authors have declared that no competing interests exist.

of healthcare workers (HCWs) reporting having experienced incivility events [3–5]. Incivility tends to be more pronounced in the surgical area [6]. For example, 53% of operating staff (nurses, surgeons, anesthetists, technicians) reported they had been subject to uncivil behavior in theater [7] and recently 92% of surgeons, surgical residents, or surgical fellows reported experiencing at least one form of incivility in the past year [5]. In a study across seven countries, nearly all respondents reported being exposed to incivility or disruptive behavior in the past year, experiencing an average of 61 incivility incidents per year [3]. Incivility can have very detrimental consequences on effectiveness, performance, patient outcomes, and employee well-being [1,8,9]. In healthcare, incivility impacts compliance with infection control and medication protocols [10], operating performance [11,12], staff well-being [13], and job attitudes [5]. Additional research is needed to understand why incivility has these consequences and mechanisms to reduce its impact.

Burnout, stress and related decrements in job attitudes, such as low satisfaction and loyalty are widespread, have been escalating among HCWs [14–16] and have substantial long-term costs due to turnover, lower productivity, and reduced patient satisfaction [17]. One source of burnout and lower well-being is incivility in teams [2,10,13]. Incivility can also lead to problems in team dynamics, such as poor communication, increased conflict, and reduced information or workload sharing [10,12,13]. Team members need to anticipate each other's actions and adjust their behavior accordingly in order to offer assistance or recognize an error and speak up [18,19]. Positive team dynamics and communication are also critically important for HCWs' well-being [20,21]. This implies that team dynamics may be a mediator in the relationship between incivility and well-being at the individual level, i.e., the relationship between incivility and well-being occurs through the impact of incivility on team dynamics.

Incivility has also been shown to impact team performance, safety, and patient outcomes [22]. For example, in a simulated operative crisis, exposure to incivility led to decreased vigilance, communication and patient management; and team members were unaware of these effects [11]. Relationships between incivility and lower safety culture have also been demonstrated [13,23]. As a result, it is imperative that these behaviors be reduced in the operating environment.

Despite documented issues of incivility, few targeted interventions have been investigated. Among those studied, CREW entails a series of workshops and training sessions over a period of weeks or months that focus on social interactions and (un)civil communication, but this intervention has demonstrated weak or inconclusive results [24]. More recent interventions have tended to focus on education and cognitive rehearsal techniques to teach HCWs (primarily nurses) how to respond to acts of incivility and build resilience [25] with some success. However, these cognitive interventions put the burden on the target of incivility rather than addressing ways to reduce incivility in the system. Other interventions have been aimed at addressing sources of patient stress such as wait times and educating patients to reduce incivility stemming from patients towards HCWs [26]. Initiatives such as the Promoting Professional Accountability Program [27] and Ethos program [28] have helped reduce descriptive behaviors. These programs focus on peers identifying and privately providing feedback about a range of behaviors (including hygiene, hand washing, patient safety and interpersonal behavior). While these programs can help reduce incivility, they are costly as successful implementation requires trained personnel to implement the surveillance system and for staff to be trained in its use [27].

To add to the growing literature on incivility in healthcare, we investigate relationships between surgical team members' experiences and observations of incivility, team dynamics, and ultimately well-being using mediated analytical models. In addition, we explore a novel intervention to reduce incivility in the operating theater environment at the aggregate level of

analysis. Because human cooperative behavior is purported to be largely regulated by social sanctions, being observed can unconsciously prompt people to modify their behavior accordingly [29]. This effect can be evoked with images of eyes that are only minimally similar to actual observer's eyes. Evoking perceptions of being observed, such as through a picture of eyes on the wall, has been shown to reduce crime and littering, and to increase honesty, voting behavior, and charitable giving [30–34]. Likewise, visual stimuli with eyes increased hand hygiene in a public restroom [35] and in a perioperative setting [36]. Theoretically, people are sensitive to subtle cues of being watched and then attempt to behave in socially acceptable ways to conform to social norms [37] or to protect against the spread of negative information that could affect their reputation [38]. Thus, even small pictures of eyes can trigger automatic cognitive mechanisms to regulate behavior. To test the impact on reducing incivility, we introduced an easy, low-cost intervention by placing 'eye' signage in operating theater areas.

## Methods

After University Research Ethics approval, surveys were distributed, and the 'eye' intervention was introduced. Participants were perioperative surgical team members (surgeons, anesthetists, assistants, ancillary specialists, scrub nurses, admission nurses, and recovery nurses) at an orthopedic surgery hospital in Australia. Surgeons are owners, managers, and workers in the hospital, making any effects of incivility highly salient. As an independent private hospital owned by surgeons, surgical staffing is stable over time. During the study period, there was almost no turnover of nursing and other staff, and anesthetists, assistants, and technicians are regular team members. There are 15 permanent surgeons, and a similar number of regular anesthetists and assistants. Regular staff also include 20 scrub/theater nurses, 30 admission/ recovery/day nurses, and 5 regular ancillary specialists/technicians with part-time staff utilized as needed. Concurrent survey methods, whereby measures were collected on the same survey, were used to examine relationships between incivility, team dynamics, and ultimately well-being outcomes of team members at the individual level. Survey data was collected at Time 1 and at Time 2, approximately 3 months apart.

The hospital Chairman, a surgeon, sent an email to participants announcing a survey would be forthcoming as part of the quality improvement initiatives at the hospital and encouraged participation in completing the surveys. Approximately one month after initial survey collection, eye signs were placed in operating theater areas, with the Time 2 survey occurring about 3 months after the initial survey. Participants were unaware of why signs were placed in theater areas, only that we were asking them to complete a short survey as part of ongoing quality assessments.

The brief surveys assessed incivility, team dynamics, burnout, stress, and job attitudes. Paper surveys and signage requesting participation were placed in breakrooms and hallways along with envelopes and a sealed box in which to place completed surveys. Participation was voluntary. Consent was indicated by completing the survey. Given the relatively small single hospital where staff know one another, in order to preserve anonymity and confidentiality in this small unit setting, no sex, demographic or background information, aside from role, was collected.

### Survey measures

The survey measured incivility, team dynamics, burnout, stress, and job attitudes. Unless otherwise noted, a 5-point scale (strongly disagree to strongly agree) was used. All items were based on previously published measures. Survey items are available at https://doi.org/10. 17605/OSF.IO/67HNV.

*Incivility* was assessed with 7 items (e.g., purposefully ignoring someone, insulting comments, speaking ill of someone, comments based on sexual or racial stereotypes, hurtful sarcasm) based on prior measures [3,39]. Participants were asked to report on a 5-point scale how often (from never to most days) they observed or experienced each event during the past month. The average score across all items (incivility events) was calculated. In addition, the number of different incivility events observed or experienced at least once in the past month was calculated.

*Team dynamics* was measured with 7 items, adapted from previously validated measures [40–42], focusing on communication, psychological safety, and respect. Sample items included, "team members feel free to raise concerns about actions or decisions," "conflicts and differences are handled privately, rather than in front of others," "members of the team respect each other's contributions," and "we work together as a well-coordinated team." *Burnout* items focused on feelings of being drained, strained, and worn out from work, taken from the Maslach and Jackson scale [43]. In order to measure *Stress*, participants were asked to indicate the amount of felt stress in the past month on a scale of 1 (none) to 10 (intense). One item measures of stress have been validated in past research [44]. Overall *Job Attitudes* included 3 items of overall feeling of satisfaction with the job, feeling a sense of loyalty to the hospital, and intention to leave within 12 months. These items have been used extensively in organizational behavior studies [45].

*Areas for Improvement* (open-ended question)–A space on the survey was provided at the end of the survey for participants to write any comments or areas for improvement.

## Eye intervention

A pretest-posttest design, at the aggregate level, was used. Approximately one month after baseline survey data on incivility was collected and signs depicting an eye(s) and a slogan were strategically placed in operating theaters and surgical hallways. Examples of the signs are depicted in Fig 1. Eye images were obtained from publicly available common stock. Slogans were added beneath the eye images and were adapted from phrases utilized in Royal Australasian College of Surgeons materials and values [46]. A total of 16 eye signs were placed, 2 in each of the 5 operating theaters. Signs were of 2 sizes (210 x 297mm and 297 x 420mm). The remaining 6 signs were placed on the walls in theater hallway areas.

Approximately seven weeks after the signs had been in place, survey data was again collected over a period of 10 days. Thus, the total time between survey administrations was about 3 months. Past research using eye signs has largely assessed the more immediate impact of eye signs on behavior such as honesty [31]. We sought to determine if the placement of signs

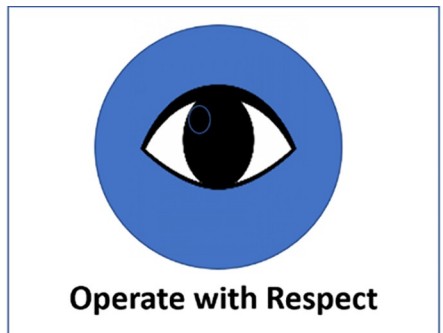
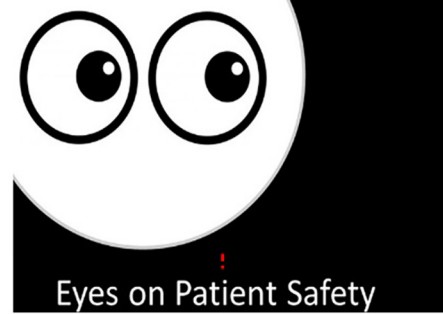

**Fig 1. Examples of eye intervention signs.**

would have an impact on behavior over a longer period of time. The timing was chosen to be long enough to assess more than an immediate, 'one-off' effect but short enough to minimize any potential effects of staff turnover (there was almost no turnover in our time period). Changes in incivility between pre-and post-intervention were examined. While an ideal design would have included random assignment of half of the surgical teams to the intervention and half as a control group, this was not feasible in a single hospital structure where team composition is fluid. In particular, scrub nurses and other staff work with multiple surgeons on a regular basis. Nevertheless, given that there were almost no changes in staff during the study period, a pre-post design allows for examining the impact of eye signs on reported observations of incivility in a group of surgical staff.

## Statistical analyses

All analyses were conducted in SPSS version 28, with $p < .05$ considered statistically significant. Mediation analyses using PROCESS [47] were performed to examine mediation of team dynamics in the relationships between individual-level incivility and each of the well-being outcomes, using 5000 bootstraps per model. ANOVA was used to test aggregate differences in incivility before and after the 'eye' intervention and by role.

## Results

A total of 74 HCWs responded to the Time 1 pre-intervention survey and 45 to the Time 2 post-intervention survey. Given the small numbers and highly similar responses, surgeons, anesthetists, and assistants were combined into one role category. The number of respondents by role is contained in Table 1. Reported incivility was similar to past research, with 93% of participants experiencing or observing at least 1 incivility event at Time 1 and 86% at Time 2. On average, they reported experiencing or observing incivility between once a week and several times per week (Mean = 2.22, $SD$ = 0.96), and 4 different types of incivility events within a month (Mean = 4.04, $SD$ = 2.29).

At the individual level, means, standard deviations and intercorrelations among incivility, well-being outcomes (burnout, stress, and job attitudes) and team dynamics are presented in Table 2. Incivility was significantly ($p < .001$) correlated with greater burnout ($r$ = .41), stress ($r$ = .34) and poorer job attitudes ($r$ = -.45).

Results of the mediation analyses indicated that the relationship between incivility and each of the well-being outcomes was statistically significant ($p < .001$) and fully mediated by team dynamics. For burnout, the overall model was significant ($R^2$ = .30) with a significant indirect effect of communication ($B$ = .26, $SE$ = .09) based on bootstrap 95% confidence intervals (lower CI = .12, upper CI = .47). Similar results were obtained for stress with an $R^2$ of .28, and an indirect communication effect ($B$ = .76, $SE$ = .28; lower CI = .35, upper CI = 1.45) and for job attitudes with an $R^2$ = .32, and indirect effect ($B$ = -.25, $SE$ = .09; lower CI = -.44, upper

**Table 1. Number of respondents by role.**

| Role | Time 1 $n$ | Time 2 $n$ |
|---|---|---|
| Surgeons, Anesthetists, Assistants | 15 | 5 |
| Other Ancillary Technicians | 14 | 8 |
| Scrub/Theater Nurses | 13 | 10 |
| Admit, Day and Recovery Nurses | 26 | 20 |
| Not Identified | 6 | 2 |
| Total | 74 | 45 |

**Table 2. Means, standard deviations and intercorrelations.**

|  | Mean | SD | Incivility | Team Dynamics | Burnout | Stress | Job Attitudes |
|---|---|---|---|---|---|---|---|
| Incivility | 2.22 | 0.96 | 0.89 |  |  |  |  |
| Team Dynamics | 3.46 | 0.72 | -0.56 | 0.83 |  |  |  |
| Burnout | 3.00 | 1.01 | 0.41 | -0.53 | 0.87 |  |  |
| Stress | 5.75 | 2.61 | 0.34 | -0.53 | 0.76 | - |  |
| Job Attitudes | 3.47 | 1.05 | -0.45 | 0.54 | -0.51 | -0.40 | 0.75 |

Note: Diagonal entries are coefficient alpha reliability.

CI = -.09). Fig 2 contains direct coefficients of the relationships. Thus, incivility has a negative association with team dynamics and communication. In turn, more positive team dynamics are associated with lower burnout and stress and with higher job attitudes.

ANOVA was used to examine the impact of the 'eye' intervention on incivility with the two main effects of job role (see Table 1) and time (pre and post-intervention). The main effects of roles ($F$ = 10.006, $p$ < .001) and time ($F$ = 4.502, $p$ = .039) showed significant differences in incivility between job roles, and a significant overall decrease in incivility from pre- to post-intervention. Further, there was a trend toward a role-by-time interaction ($F$ = 2.56, $p$ = .08) driven primarily by reductions in incivility reported by scrub/theater nurses and surgeons/anesthetists/assistants. Findings by role are depicted in Fig 3.

In order to provide additional depth and support for the findings and to identify any additional insights, we provided space on the survey for participants to provide comments to a broad question of areas for improvement. 47% of survey respondents provided comments. Following standard processes for thematic analysis [48], one of the study authors became familiar with the comments, identified broad code areas from them, coded each comment into one of the theme areas, revising and adding themes along the way, and finally collating themes and comments. A second author then reviewed all the themes and coding, and the two researchers decided to collapse some themes together given substantial overlap. Thematic analysis grouped responses are presented in Table 3, with representative comments to illustrate each theme.

Four themes were identified: 1) management practices and behaviors, emphasizing the need for higher level management to be role models and emphasize respect in the culture, 2) employee appreciation, highlighting the importance of acknowledging that people are valued, 3) communication, focusing on open exchanges within teams and between employees and management; and 4) work practices, emphasizing the importance of role clarity. The comments reflect the premises of the study about the role of incivility in workplace culture, team dynamics, and well-being.

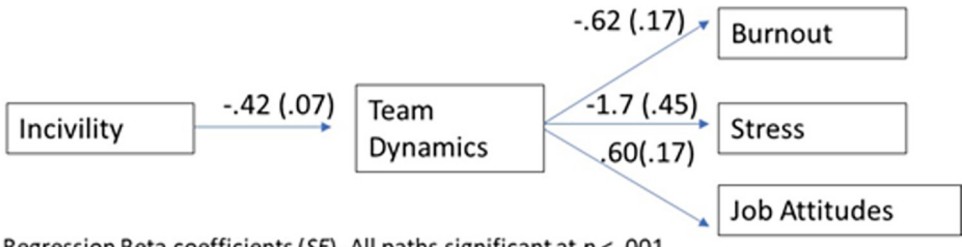

Regression Beta coefficients (SE). All paths significant at $p$ < .001

**Fig 2. Relationship between incivility and outcomes mediated by team dynamics.**

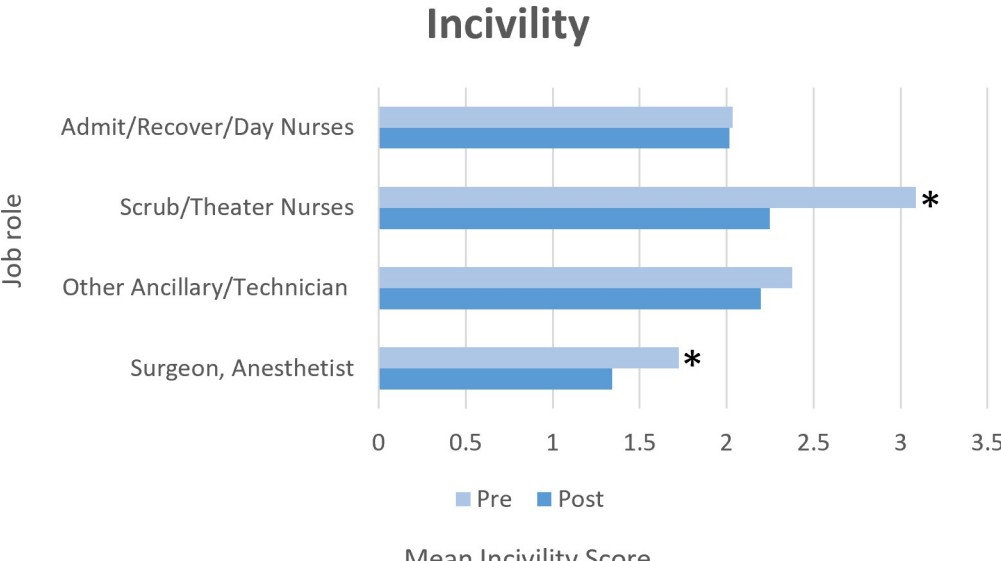

**Fig 3. Pre- and post-intervention incivility by job roles.**

## Discussion

In this study, reports of incivility were in line with those reported in other institutions worldwide [3–7]. While the absolute frequency may appear small (i.e., incivility events were observed or experienced a few times a week on average per person), efforts to reduce incivility are warranted given the growing evidence that incivility has the potential to impede team coordination and impact safety and patient outcomes.

Incivility can have significant costs to the well-being of staff and ultimately to patients, as well as having a negative impact on the overall culture, work performance, absenteeism and turnover, and commitment to the organization [5]. While research is beginning to address detrimental impacts of incivility, there is a paucity of work explicating mechanisms through which incivility can impact well-being outcomes. Our study demonstrated that those who report observing or experiencing greater incivility have higher burnout, stress, and lower job attitudes (satisfaction, loyalty and intention to leave). This highlights that incivility in the surgical culture and climate is a risk factor for well-being, job attitudes and intentions to leave the job.

Importantly, using mediation analytical techniques, we were able to show that the relationship between incivility and well-being in surgical teams occurs largely through its impact on team dynamics, i.e., constructive communication in a psychologically safe manner. This is important because communication and teamwork are critical elements for safety and optimizing care [20] and a substantial percentage of preventable adverse effects in surgery can be traced to communication failures [21,49]. Of note, there can be significantly different perceptions of communication among surgical team members, with surgeons tending to report better communication compared to those in other roles [50–52]. Likewise, growing evidence highlights differences between male and female surgeons in interpersonal communication and conflict [53]. An ethnographic study of surgical team communication patterns revealed different types of communication and relational patterns (e.g., proactive, silent/ordinary, inattentive/ambiguous, contradictory) [54] which presumably might be impacted by the degree of incivility. While we did not have sufficient power to delve into these nuances, the intersection of roles, incivility, differing perceptions of team dynamics and communication and means for

**Table 3. Themes and sample quotes from open-ended responses.**

| Theme | Description | Supporting Quotations |
|---|---|---|
| Management Practices and Behaviors | Senior management engagement with employees, seeking input from employees, and role modeling appropriate behaviors. | "The surgeons at the 'top' mostly treat their staff with disrespect...If the surgeons were positive role models who showed respect and kindness, it would create a positive environment to work in from the top down."<br>[Managers should] "Act more professionally and not bad mouth employees in front of other employees."<br>"Management need to listen to those on the working floor."<br>"Management need to change their attitude towards staff."<br>"Surgeons and upper management need to be a role model for positive behaviour." |
| Employee Appreciation | Expressing gratitude and care towards employees. | [Senior management should]<br>"Acknowledge staff when they have gone above and beyond. Thank them."<br>"Listen and act on concerns raised by employees. To feel like higher management actually care."<br>"Take note when staff morale is down and thank people for being here, doing their jobs well everyday." |
| Communication | Openness and information sharing among employees and with management. | "If issues have occurred, [management should] address the people concerned with a genuine conversation. Work as a team, be respectful, mindful and stop the bullying attitudes."<br>"To have better communication between exec management and surgeons to the staff as they always say they are working on making it better but it never happens."<br>"People whose performance requires oversight/discussion should be spoken to–rather than about (to other people) in order to solve the problem for staff affected."<br>"I feel that some members of the team do not understand my role. They therefore continue to expect more than I am physically able to cope with. They could possibly be more involved in providing support to enable me to manage my role better."<br>"I think if all the negative talk stopped and more positive talk might up-lift the rest of the team."<br>"Still a feeling that you voice your opinions to OR's that they will not be heard. Nurses & doctors do not trust each other's abilities and second guess decisions resulting in things slowing down." |
| Work Practices | Degree of role clarity, salient processes, and accountability. | "The input and roles of management team are less clear and effective. Seems to be many roles at that level and no real input that has satisfactory outcomes."<br>[We need to have] "Clear and concise rules + templates to follow for all practices so everyone is on the same page and exec team and management can back staff decision making based on these templates and rules."<br>"Processes need to have a clear 'owner' and management need to back up staff when we say this is how it is done per policy so as management we will back you up and make surgeons stick to this policy." |

addressing them is an area ripe for more investigation. To the extent that incivility hinders communication, patient outcomes can be compromised putting a spotlight on finding ways to reduce incivility. Our findings point to another reason why additional interventions to increase teamwork dynamics need to be explored in surgical teams.

A primary purpose of the study was to explore whether incivility can be easily altered, as a way to further highlight the issue and spur future research. Our study demonstrated that uncivil behaviors have the potential to be modified, which is important for future work on tackling this issue. The simple intervention of placing 'eye' signs in the theater areas had a significant impact on reducing incivility events, particularly for theater scrub nurses. Given that participants were unaware of the purpose of the study, the finding that incivility was reduced over two months suggests the impact was largely unconscious in modifying behavior. This is consistent with the premises in prior work demonstrating that monitored observation alters behavior in socially desirable ways, with the effect so strong that even depictions of eyes can unconsciously evoke a sense of being observed [31].

While effective in our study, it is unlikely that the effect of an intervention such as 'eye' signs will be long-lasting. Thus, it should be viewed as one component of a larger program to address and reduce incivility and additional interventions should be explored to complement and extend the findings for longer-term culture change. For example, implementing new systems for staff to report incivility without fear of retribution or judgment can enhance well-being [55,56]. Additional research is needed into mechanisms that address facilitating communication and getting people on the same page about team dynamics. For example, implementing pre-surgery briefings and post-surgery debriefings processes can facilitate better team communication [57] and it is likely these mechanisms could also help reduce incivility. Identifying specific sources of incivility so they can be addressed is also useful and worthy of future research attention. For example, high workload and poor coordination problems are often primary triggers of disruptive behaviors [58]. More generally, wide-sweeping initiatives that focus on peer monitoring of behaviors, such as the Promoting Professional Accountability Program which has been implemented in countries such as the US, Australia, Singapore and the UK [27] and the Ethos program in Australia [28], have had an impact on reducing a range of unprofessional behaviors and improving safety. Long-lasting and sustained reduction in incivility is likely to require such system-wide and comprehensive approaches with multiple components that result in broader culture change. The open-ended comments reflect the notion that respectful and professional treatment is critical. Further, they draw attention to the primary influence of the broader culture and climate, as well as higher-level leadership [59] which mitigate incivility issues [58]. The comments echoed the importance of developing a culture and climate that values and appreciates employees, shows compassion, engages staff, and facilitates communication, with leadership and role modeling as drivers of these systems [60].

Limitations of the study include the cross-sectional nature of the individual-level survey, reliance on self-reports of incivility, the small sample size, reduced sample size from Time 1 to Time 2, and implementation in a single small hospital. Further, the study was conducted during the Covid-19 pandemic during a time of uncertainty, pressure and work demands. While this could have resulted in elevated levels of stress, burnout, or incivility overall, it does not negate the significant relationships we demonstrated between them, which are theoretically consistent with the literature. Nor should the timing of the study during the pandemic impact whether signs would reduce incivility or not. Additional research using other methods such as behavioral observation of incivility and teamwork, longitudinal individual and team data collection and larger cross-hospital samples would be useful to enhance the findings of this study. Despite the limitations, the findings point to the importance of future research into the mechanisms by which incivility has detrimental impacts and the investigation of additional interventions and processes which can create more civil team dynamics.

## Author Contributions

**Conceptualization:** Cheri Ostroff.

**Data curation:** Belinda Rae, Douglas Fahlbusch, Nicholas Wallwork.

**Formal analysis:** Cheri Ostroff.

**Funding acquisition:** Cheri Ostroff.

**Methodology:** Cheri Ostroff, Chelsea Benincasa, Belinda Rae.

**Project administration:** Cheri Ostroff, Chelsea Benincasa, Belinda Rae, Nicholas Wallwork.

**Writing – original draft:** Cheri Ostroff.

**Writing – review & editing:** Cheri Ostroff, Chelsea Benincasa, Belinda Rae, Douglas Fahlbusch, Nicholas Wallwork.

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
