## [Decision Letter · Decision Letter 0]

20 Jul 2023

PONE-D-23-13370Eyes on Incivility in Surgical Teams: Teamwork, Well-Being, and an InterventionPLOS ONE

Dear Dr. Ostroff,

Thank you for submitting your manuscript to PLOS ONE. After careful consideration, we feel that it has merit but does not fully meet PLOS ONE’s publication criteria as it currently stands. Therefore, we invite you to submit a revised version of the manuscript that addresses the points raised during the review process.

I would like to sincerely apologize for the delay you have incurred with your submission. It has been exceptionally difficult to secure reviewers to evaluate your study. We have now received three completed reviews; the comments are available below. The reviewers have raised significant scientific concerns about the study that need to be addressed in a revision.

Please revise the manuscript to address all the reviewer's comments in a point-by-point response in order to ensure it is meeting the journal's publication criteria. Please note that the revised manuscript will need to undergo further review, we thus cannot at this point anticipate the outcome of the evaluation process.

We look forward to receiving your revised manuscript.

Kind regards,

Miquel Vall-llosera Camps

Senior Editor

PLOS ONE

Journal Requirements:

Reviewers' comments:

Reviewer's Responses to Questions

**Comments to the Author**

1. Is the manuscript technically sound, and do the data support the conclusions?

Reviewer #1: Yes

Reviewer #2: Yes

Reviewer #3: Partly

2. Has the statistical analysis been performed appropriately and rigorously? 

Reviewer #1: Yes

Reviewer #2: Yes

Reviewer #3: I Don't Know

3. Have the authors made all data underlying the findings in their manuscript fully available?

Reviewer #1: Yes

Reviewer #2: Yes

Reviewer #3: Yes

4. Is the manuscript presented in an intelligible fashion and written in standard English?

Reviewer #1: Yes

Reviewer #2: Yes

Reviewer #3: Yes

5. Review Comments to the Author

Reviewer #1: The authors examined the effectiveness of an intervention (posters with eyes and messages) at reducing incivility in operating theatre staff. The study included a pre and post survey. Key findings include reported incivility was lower at follow up and those who reported experiencing incivility had higher burnout, stress, and lower job attitudes.

The Methods, Results, and Discussion sections require additional information for clarity. These sections would need to be strengthened if the manuscript is to be considered for publication. A significant limitation is the design of the study with the short time from intervention to follow-up and the small number of staff responses. However, given the scarcity of studies that have examined the effectiveness of interventions to reduce incivility in hospitals, this study still has value and is an important contribution to the literature.

The authors should be commended as it is an interesting and important study that adds further to the development of interventions designed to reduce incivility in hospitals. I have included suggestions for minor revisions. There are also some minor grammatical and typographical errors throughout the manuscript.

Abstract

“Participants responded to an open-ended question about suggestions for improvements that was thematically analyzed.” Please clarify when participants responded to this question.

Introduction

The first sentence of the third paragraph is out of place. Consider removing.

The final sentence of the third paragraph is out of place. Consider removing.

Materials and Methods

Please include the survey items as an appendix.

Were the eye designs adapted from existing material? How were they designed?

Please provide an appropriate citation for the phrases from the Royal Australasian College of Surgeons. And change “Australian” to “Australasian”.

In the second paragraph of this section, it is stated that “Survey data was collected at Time 1 and Time 2, approximately 3 months later.” This is confusing because later in the eye intervention sub section it states that baseline data were collected, one month later the signs were posted, and then 1 month after that data were again collected. Is this not 2 months? Please clarify.

It is unclear why follow-up data were collected 1-month post intervention. Why not 2 or 3 months? Please explain.

Please include details of how many operating theatres were at this site. How many posters were placed in the operating theatres and how many were placed in the surgical hallways? Where exactly were they placed in operating theatres? These are important details.

Did you measure at post intervention whether staff actually saw the signs?

There is no mention of the open-ended question about suggestions for improvements in the Materials and Methods section (it is mentioned in the Abstract). Please add.

In the Statistical analyses subsection, please add a description of your approach to the thematic analysis. Also, what software did you use?

Results

What were the total numbers for each staff role at the hospital? What were the response rates for each role given the total staff?

Can you add a row that includes the totals for each timepoint for Table 1?

It is stated that 91% of respondents experienced at least 1 incivility event at Time 1. What about Time 2? Please add.

In the 4th paragraph, please include the actual p value rather than writing p<.05.

Can you please label the x axis for Figure 3?

How many staff responded to the open-ended question?

Would it be possible to include the staff role with the quotes in Table 3?

Discussion

The Discussion section does not clearly explain how the intervention could have reduced incivility. For example, “Given that participants were unaware of the purpose of the study and the signs…”, but surely staff would have to be aware of the signs if the signs are to have an effect. The authors need to clarify this and draw on existing literature to explain how this effect could happen.

The authors have neglected to mention the work done in the US (Promoting Professional Accountability Program - Vanderbilt University Medical Centre) and in Australia (Ethos program) where professional accountability culture change programs have been developed to address unprofessional behaviour in hospitals. This should be added to both the Introduction and Discussion.

Reviewer #2: Thank you so much for the opportunity to review this very relevant, innovative, and inspiring paper focusing on the relationships between incivility, team dynamics and well-being in surgical teams. It was very interesting to read the well-argued and rigorous introduction describing why it matters to be attentive to incivility, how it negatively impacts staff well-being and effectiveness and why a low-cost intervention was chosen. An interesting description of materials and design, results and suggestions for further exploration. Please find below some comments to improve the paper.

Abstract & Introduction: In the abstract, the authors state the purpose and summarize the main research question and key findings. The Introduction chapter is clear, well-argued, and has an extensive use of literature on the topic.

Materials and Methods: In this chapter ethical issues are well described in detail. The materials and survey measures are clearly described. Burnout items are measured using the Maslach and Jackson scale.

Page 6: The items measuring Team dynamics are adapted from previously validates measures…. I am curious about these items, and it is not clear for me when reading the references (35, 36, 37). It would improve the paper to be more specific about these items or validated measurement tools for measuring team dynamics, and maybe also be clearer about how the three items for Job attitudes are emphasized and how they are formulated,

Page 7: Statistical analyses are well described except for the thematical analysis of the open-ended questions.

Results: A content-rich and clearly described chapter

Page 7-8: Comments on Table 1. The authors may consider writing the same names of job functions in Table 1 as in the Materials section (Page 5). Ancillary specialist is it the same as Other Ancillary Technicians? Is Scrub Nurses the same as Theatre Nurses? Or why are Theatre Nurses not mentioned in Materials (Page 5). Who are Day Nurses? And why are they not mentioned in Materials (Page 5)?

Page 8: Comments on Table 2. The authors may consider writing the same names on the items. Team Dynamics in the vertical column and Team in the horizontal row. Job Attitudes in the vertical column and Attitudes in the horizontal row. I assume it is the same item. Perhaps the full name is hidden in the Table?

Page 9: Comments on the thematical analysis. The authors might consider adding some comments on how the identification was conducted.

Discussion: An interesting discussion highlighting the findings and the value of these for future studies on how to reduce incivility and thereby improve team dynamics in the surgical teams – “that uncivil behaviors have the potential to be modified”.

The authors may consider including a comment on the fact that the percentage response from the group of Admit, Day and Recovery nurses is considerably greater Time 2 (20 out of 45) than Time 1 (26 out of 74). How does it affect the results in the light of the small difference between the Pre and Post measurements in this group (Fig 3)? And maybe a comment on the significantly reduced response from the group of Surgeons, Anesthetists and Assistants Time 2.

Finally, the authors might consider adding some comments on how the open-ended responses could be used in order to improve the communication in the surgical teams.

Reviewer #3: This manuscript targets an enduring problem in the OR, namely incivility and its impacts on participants and patient care, and it does so through a novel and simple intervention. While I read the manuscript with interest, there were sections that seemed to lack depth according to my reading, which affects the perceived strength of the findings and conclusions. Please consider the following observations

:

Introduction: some of the claims in the background section are not well supported by the references supplied e.g.. "53% of operating staff'..had reported... in theatre. The citation is 15years old and is published in a nursing journal (so may reflect only nurses' perspective about others in the OR team'. More context from they study is needed to support the claim as well as a more contemporary reference.

- Introduction; there is very limited if any background on existing interventions. As the manuscript will be of interest to health professions educators and communication researchers, some contextualising of interventions for the OR and the role of simulation in education could enhance this section e.g. Rogers et al below, or see work of Jenny Weller, A Merry from NZ.

- Introduction some concepts might not be accessible to interdisicplinary readers and could benefit with a brief gloss e.g. 'lower team dynamics', mediation analytical techniques (discussion), concurrent survey methods.

Methods:

context - were the participants likely to be from OR teams that regularly worked with each other , literature suggests that teams that are dynamic and changing (in terms of constitution) can contribute to the communication load?

- concerning that not even Sex was collected in the demographic data as there is literature on incivility through a gender lens (Dossett for example see below). Suggest add this to limitations.

- there is very little information on the intervention E.g. what were the slogans used ?

- some explanation of the free text analysis is needed, and how the themes were arrived at is not clear.

What was the size of the pool of OR staff that the participants were recruited from.

Results.

suggest the examples of quotes from communication, pasted below, are more about work practices and roles than communication

“People whose performance requires oversight/discussion should be spoken to – rather than about (to other people) in order to solve the problem for staff affected.”

“I feel that some members of the team do not understand my role. They therefore continue to expect more than I am physically able to cope with. They could possibly be more involved in providing support to enable me to manage my role better.”

typos and stylistic aspects:

- abstract: '....to evoke perceptions of being observed' should come earlier when 'simple intervention is first mentioned'.

- in the abstract discussion section 'a simple intervention such as signage etc 'could be rephrased to reduce the repetition.

- the Lingard et al 2004 reference is repeated (refs 21 and 42).

Refs suggested follow up.

] Dossett LA, Vitous CA, Lindquist K, et al. Women surgeons‘ experiences of interprofessional workplace conflict. JAMA Network Open. 2020;3(10):e2019843– e2019843.

[33] Rogers D, Lingard L, Boehler ML, et al. Teaching operating room conflict management to surgeons: clarifying the optimal approach. Med Educ. 2011;45 (9):939–945.

[34] Grade MM, Tamboli MK, Merrell SB, et al. Attending surgeons differ from other team members in their perceptions of operating room communication. J Surg Res. 2019;235:105–112.

6. PLOS authors have the option to publish the peer review history of their article (what does this mean?). If published, this will include your full peer review and any attached files.

Reviewer #1: No

Reviewer #2: No

Reviewer #3: **Yes: **Robyn Woodward-Kron

---

## [Author Response · Author response to Decision Letter 0]

23 Aug 2023

5. Review Comments to the Author

Reviewer #1: The authors examined the effectiveness of an intervention (posters with eyes and messages) at reducing incivility in operating theatre staff. The study included a pre and post survey. Key findings include reported incivility was lower at follow up and those who reported experiencing incivility had higher burnout, stress, and lower job attitudes.

The Methods, Results, and Discussion sections require additional information for clarity. These sections would need to be strengthened if the manuscript is to be considered for publication. A significant limitation is the design of the study with the short time from intervention to follow-up and the small number of staff responses. However, given the scarcity of studies that have examined the effectiveness of interventions to reduce incivility in hospitals, this study still has value and is an important contribution to the literature.

The authors should be commended as it is an interesting and important study that adds further to the development of interventions designed to reduce incivility in hospitals. I have included suggestions for minor revisions. There are also some minor grammatical and typographical errors throughout the manuscript.

We appreciate your positive and constructive comments. 

We agree that more details and clarity are needed and have attended to this throughout the revision (specifics are outlined below). We also agree that there is a dearth of intervention strategies, particularly in surgery, and those that have been done with HCWs have tended to focus either on a) educating and training people to better respond and have more resilience when faced with incivility rather than reduce systemic incivility, and/or b) focused on reducing patient incivility towards HCWs. This is clarified in the introduction. We also agree there are some limitations to the study design, which we note in the discussion, and highlight that our study is a demonstration that incivility can be altered and more research is needed to develop lasting interventions.

Abstract

“Participants responded to an open-ended question about suggestions for improvements that was thematically analyzed.” Please clarify when participants responded to this question.

Thank you for pointing out the lack of sufficient detail. We have clarified this in the abstract as well as the text (pg. 7). In the abstract, we now state: “Participants responded to an open-ended question about suggestions for improvements at the end of the survey that was then thematically analyzed.”

On pg. 7, we amended the sentence to state: “A space on the survey was provided at the end of the survey for participants to write any comments or areas for improvement.”

Introduction

The first sentence of the third paragraph is out of place. Consider removing.

The final sentence of the third paragraph is out of place. Consider removing.

Thank you for these suggestions. We have deleted the two sentences as you suggest. And we added clarity at the end of the paragraph to emphasize that we examine mediated relationships (incivility to team dynamics to well-being) at the individual level of analysis. 

Materials and Methods

Please include the survey items as an appendix.

As our items are all derived from previously used measures, we did not include them in an appendix because we did not feel it would be a good use of journal space. In response to this comment (and reviewer 2), we have added more details in the methods and more sample items, and all of the original source references are included. 

If the editor would like us to include the full survey, we will be happy to do so.

Were the eye designs adapted from existing material? How were they designed?

Based on eye signs used in past research in other areas, we focused on depictions (illustrations) of eyes. We used publicly available common stock pictures of eyes. To these, we added slogans and language from the RACS (Royal Australasian College of Surgeons). Signs were placed in theatres and theatre hallways. We have added more details to clarify this in the methods section on pg. 8.

Please provide an appropriate citation for the phrases from the Royal Australasian College of Surgeons. And change “Australian” to “Australasian”.

We corrected this typo, and added a reference to RACS. 

In the second paragraph of this section, it is stated that “Survey data was collected at Time 1 and Time 2, approximately 3 months later.” This is confusing because later in the eye intervention sub section it states that baseline data were collected, one month later the signs were posted, and then 1 month after that data were again collected. Is this not 2 months? Please clarify.

Thank you. We agree this was confusing. Survey data was collected at Time 1 over a period of 10 days. One month later, signs were posted in the operating theatres and hallways outside the theatres. 7 weeks after signs were placed, Time 2 survey data collection began over a period of 10 days. This has been clarified on page 8.

It is unclear why follow-up data were collected 1-month post intervention. Why not 2 or 3 months? Please explain.

This is a good question. As noted above, we clarified that signs were placed approx. 1 month after the pre survey, and then the second survey data collection occurred approximately 7 weeks later. There is no research to indicate the ideal or most appropriate time frame for an ‘eye’ intervention. The preponderance of past research using eye signs (on behaviors such as honesty or theft) has been to examine the effect on immediate behavior. Rather than a one-time, one-shot test, we aimed to determine if our simple intervention could have a demonstratable effect for weeks, showing effects for close to 2 months after being introduced. Further, we wanted a gap that was long enough to examine more than an immediate effect, but short enough to minimize any potential effects of staff turnover if a longer timeframe were used. Clearly future research is needed to examine nuances of the impact of signage, length of the effect, and the like, as well as different additional systemic interventions. We clarify this in the methods (pg.. 8) and note the need for additional interventions in the discussion.

Please include details of how many operating theatres were at this site. How many posters were placed in the operating theatres and how many were placed in the surgical hallways? Where exactly were they placed in operating theatres? These are important details.

A total of 16 posters were placed. 2 in each of the 5 operating theatres. Signs were 2 sizes (210x297 cm and 297x420cm). The remaining 6 signs were placed on the walls in theatre hallways and spaces. In response to your comment and that of reviewer 3, we have added more details about the signs (see pg. 8). 

Did you measure at post intervention whether staff actually saw the signs?

We did not ask whether staff noticed the signs because the purpose of the intervention was to determine if ‘eyes watching’ evoke a potentially unconscious effect on behavior. 

There is no mention of the open-ended question about suggestions for improvements in the Materials and Methods section (it is mentioned in the Abstract). Please add.

Thank you for pointing out this oversight on our part. We have clarified this in the methods (pg. 7).

In the Statistical analyses subsection, please add a description of your approach to the thematic analysis. Also, what software did you use?

We have added additional details in response to your comment. Given the purpose or providing supportive information and more depth for the quantitative aspects coupled with the fact that there was only one open ended question and a relatively small number of comments (e.g., 35 people commented on 74 surveys), we did not need to use software such as Nvivo for the thematic analysis. We followed standard processes for coding and organizing comments into themes, confirmed by a second researcher. We have outlined this on pg. 11. 

Results

What were the total numbers for each staff role at the hospital? What were the response rates for each role given the total staff?

Exact staffing numbers are complex given some part-time work in some roles. However, there are 15 permanent surgeons and similar number of regular anaesthetists and 10 assistants, 20 scrub nurses, and 30 anaesthetic/day nurses. There was almost no turnover during the study period. These details are now included on pg.s5 and 6.

Can you add a row that includes the totals for each timepoint for Table 1?

We have added this.

It is stated that 91% of respondents experienced at least 1 incivility event at Time 1. What about Time 2? Please add.

We added that 86% experienced at least 1 event at Time 2.

In the 4th paragraph, please include the actual p value rather than writing p<.05.

We now report the p value (.039), pg. 10. 

Can you please label the x axis for Figure 3?

We apologize for the oversight and have added.

How many staff responded to the open-ended question?

47%, which we now include in the methods, pg. 11.

Would it be possible to include the staff role with the quotes in Table 3?

We did not indicate role by the comment in order to help maintain some anonymity, to the extent possible, given that this is a fairly small group that knows one another. Further, we did not identify any notable trends in the comments based on role. 

Discussion

The Discussion section does not clearly explain how the intervention could have reduced incivility. For example, “Given that participants were unaware of the purpose of the study and the signs…”, but surely staff would have to be aware of the signs if the signs are to have an effect. The authors need to clarify this and draw on existing literature to explain how this effect could happen.

We agree this was not explained as well as it could have been. Participants were not aware of the purpose of the study or signs, but were asked to complete surveys as an ongoing aspect of their quality assessments. Thus, they may have noticed the signs but were unaware of their underlying purpose. Based on considerable past research examining the impact of ‘eyes,’ the mechanism is deemed to be an unconscious one. It is well known that people amend their behavior, often in more socially desirable ways, when they believe they are being watched or monitored by other individuals. This effect can carry-over to signs, whereby just an image of an eye can unconsciously make people feel they are being observed, and hence they alter their behavior. We have now added additional text in the discussion, pg. 14. 

The authors have neglected to mention the work done in the US (Promoting Professional Accountability Program - Vanderbilt University Medical Centre) and in Australia (Ethos program) where professional accountability culture change programs have been developed to address unprofessional behaviour in hospitals. This should be added to both the Introduction and Discussion.

Your point is well taken. The PPA program is used in many hospitals in the US and in several hospitals in Australia, and Ethos is very similar. These programs go beyond incivility, and include all manner of safety behaviors (e.g., hygiene/hand washing, patient safety), with clinicians encouraged to discuss ‘complaints’ with their fellow HCW. This is a larger culture change system that can help with incivility as well as a wide range of other behaviors. Thus, we did not initially include this as it was not specific to incivility per se. Given your comment, we have reconsidered its relevance and have now included links to broader culture change initiatives 

in the discussion, pg. 15. 

Reviewer #2: Thank you so much for the opportunity to review this very relevant, innovative, and inspiring paper focusing on the relationships between incivility, team dynamics and well-being in surgical teams. It was very interesting to read the well-argued and rigorous introduction describing why it matters to be attentive to incivility, how it negatively impacts staff well-being and effectiveness and why a low-cost intervention was chosen. An interesting description of materials and design, results and suggestions for further exploration. Please find below some comments to improve the paper.

Thank you. We appreciate the positive comments.

Abstract & Introduction: In the abstract, the authors state the purpose and summarize the main research question and key findings. The Introduction chapter is clear, well-argued, and has an extensive use of literature on the topic.

Materials and Methods: In this chapter ethical issues are well described in detail. The materials and survey measures are clearly described. Burnout items are measured using the Maslach and Jackson scale.

Page 6: The items measuring Team dynamics are adapted from previously validates measures…. I am curious about these items, and it is not clear for me when reading the references (35, 36, 37). It would improve the paper to be more specific about these items or validated measurement tools for measuring team dynamics, and maybe also be clearer about how the three items for Job attitudes are emphasized and how they are formulated,

We agree that additional details and sample items would be useful and have included them on pg. 7. 

Page 7: Statistical analyses are well described except for the thematical analysis of the open-ended questions.

Thank you for pointing this out. We have now included further details about the thematic analysis, beginning pg. 11.

Results: A content-rich and clearly described chapter

Page 7-8: Comments on Table 1. The authors may consider writing the same names of job functions in Table 1 as in the Materials section (Page 5). Ancillary specialist is it the same as Other Ancillary Technicians? Is Scrub Nurses the same as Theatre Nurses? Or why are Theatre Nurses not mentioned in Materials (Page 5). Who are Day Nurses? And why are they not mentioned in Materials (Page 5)?

Apologies for the inconsistencies in terminology. We have now included them earlier in the methods section and have ensured consistent terms have been used throughout.

Page 8: Comments on Table 2. The authors may consider writing the same names on the items. Team Dynamics in the vertical column and Team in the horizontal row. Job Attitudes in the vertical column and Attitudes in the horizontal row. I assume it is the same item. Perhaps the full name is hidden in the Table?

Thank you for the comment. We have expanded the labels in the Table 2 to enhance clarity.

Page 9: Comments on the thematical analysis. The authors might consider adding some comments on how the identification was conducted.

We agree. We now indicate that we followed standard procedures for thematic analysis and have included additional details on pg. 11.

Discussion: An interesting discussion highlighting the findings and the value of these for future studies on how to reduce incivility and thereby improve team dynamics in the surgical teams – “that uncivil behaviors have the potential to be modified”.

The authors may consider including a comment on the fact that the percentage response from the group of Admit, Day and Recovery nurses is considerably greater Time 2 (20 out of 45) than Time 1 (26 out of 74). How does it affect the results in the light of the small difference between the Pre and Post measurements in this group (Fig 3)? And maybe a comment on the significantly reduced response from the group of Surgeons, Anesthetists and Assistants Time 2.

Thank you for these good points. We have better addressed some of these limitations in the discussion.

Finally, the authors might consider adding some comments on how the open-ended responses could be used in order to improve the communication in the surgical teams.

We agree. We now note in the discussion that the open-ended comments highlight that incivility likely stems from a multitude of individual and broader system factors. We then link this to a discussion of wider system interventions (highlighted by reviewer 1.

Reviewer #3: This manuscript targets an enduring problem in the OR, namely incivility and its impacts on participants and patient care, and it does so through a novel and simple intervention. While I read the manuscript with interest, there were sections that seemed to lack depth according to my reading, which affects the perceived strength of the findings and conclusions. Please consider the following observations

:

Introduction: some of the claims in the background section are not well supported by the references supplied e.g.. "53% of operating staff'..had reported... in theatre. The citation is 15years old and is published in a nursing journal (so may reflect only nurses' perspective about others in the OR team'. More context from they study is needed to support the claim as well as a more contemporary reference.

We appreciate your attention to details and the timing of the data. We have provided additional data from published papers on incivility. In the original version, our first 6 references on incivility were all from 2020 onwards. Together, we believe they provide compelling evidence of the problem of incivility, and the older reference helped to show that the this has been ongoing. With respect to the citation you highlight, it was not only nurses; respondents were nurses, surgeons, anesthetists, and technical practitioner/specialists. In response to your comment, we have clarified this and provide additional details and current references to support the claim that incivility is prevalent.

- Introduction; there is very limited if any background on existing interventions. As the manuscript will be of interest to health professions educators and communication researchers, some contextualising of interventions for the OR and the role of simulation in education could enhance this section e.g. Rogers et al below, or see work of Jenny Weller, A Merry from NZ.

Thank you for the insightful comment. There are numerous factors that impact team dynamics, communication, and conflict in surgical teams. And indeed, there are several studies highlighting that surgeons often have more positive perceptions about the team dynamics than do nurses and other staff. We added the reference you note (in addition to those we had originally cited) and highlighted the need for more attention to this issue. Incivility, in our view, is only factor that contributes to lower team dynamics. In the introduction, we now cite and explain some of the incivility interventions that been examined in past research, with modest or little success (pg. 4). Further, in response to this comment, we now include broader issues about conflict management and differing perceptions of communication that merit attention in this arena for future research (pg. 13).

- Introduction some concepts might not be accessible to interdisicplinary readers and could benefit with a brief gloss e.g. 'lower team dynamics', mediation analytical techniques (discussion), concurrent survey methods.

Thank you. We have included more definitions throughout the paper for social science and analytical terms.

Methods:

context - were the participants likely to be from OR teams that regularly worked with each other, literature suggests that teams that are dynamic and changing (in terms of constitution) can contribute to the communication load?

Your point is well taken. In this case, staff did regularly work with one another. We have clarified this in the text (pg.s 5 and6): As an independent private hospital owned by surgeons, surgical staff is stable over time. During the study period, there was almost no turnover of surgical nursing staff and anesthetists and other staff are regular team members.

- concerning that not even Sex was collected in the demographic data as there is literature on incivility through a gender lens (Dossett for example see below). Suggest add this to limitations.

Again, your point is well taken. In this case, as a surgeon-owned relatively small specialist hospital, the staff know one another well and regularly work with one another. Hence, there were significant concerns about maintaining anonymity if we asked any sex or demographic information. In this setting, the large majority of surgeons were male, and the large majority of scrub nurses were female (similar to demographics in this specialty and industry). We agree that there are often sex differences reported in incivility and experiences. We were not able to investigate this, nor would we have likely had sufficient power to detect sex by role differences. As suggested, we mention this as a limitation and direction for future work in the discussion section beginning pg. 13. 

- there is very little information on the intervention E.g. what were the slogans used ?

We agree and apologize for the oversight. We have provided more details about the signs and slogans on pg. 8 with examples in Figure 1. 

- some explanation of the free text analysis is needed, and how the themes were arrived at is not clear.

We agree that we did not provide sufficient details and we have added more information on pg. 11.

What was the size of the pool of OR staff that the participants were recruited from.

Again, we agree that it would be helpful to provide more information about the staff. We have now included this information on pg. 6.

Results.

suggest the examples of quotes from communication, pasted below, are more about work practices and roles than communication

“People whose performance requires oversight/discussion should be spoken to – rather than about (to other people) in order to solve the problem for staff affected.”

“I feel that some members of the team do not understand my role. They therefore continue to expect more than I am physically able to cope with. They could possibly be more involved in providing support to enable me to manage my role better.”

This is an interesting comment. We categorized these under the broad theme of communication based on organizational psychology dimensions, where work practices are specific to practices (e.g., HR practices), policies, and procedures. Role clarity is considered a part of practices when it is about work structure and defined practiced from the top. As these issues are about understanding the roles of people within the team (due to lack of information sharing and communication), we believe they are more relevant to communication issues in the team. Of course, there is naturally overlap between practices and communication in most organizational settings. Two of the researchers independently reviewed the themes and categorization – but as we are aware, thematic analysis relies on the researchers’ experience with the data. 

typos and stylistic aspects:

- abstract: '....to evoke perceptions of being observed' should come earlier when 'simple intervention is first mentioned'.

- in the abstract discussion section 'a simple intervention such as signage etc 'could be rephrased to reduce the repetition.

We appreciate these suggestions. We have amended this wording in the abstract. 

- the Lingard et al 2004 reference is repeated (refs 21 and 42).

Thank you for noticing this. 

Refs suggested follow up.

] Dossett LA, Vitous CA, Lindquist K, et al. Women surgeons‘ experiences of interprofessional workplace conflict. JAMA Network Open. 2020;3(10):e2019843– e2019843.

[33] Rogers D, Lingard L, Boehler ML, et al. Teaching operating room conflict management to surgeons: clarifying the optimal approach. Med Educ. 2011;45 (9):939–945.

[34] Grade MM, Tamboli MK, Merrell SB, et al. Attending surgeons differ from other team members in their perceptions of operating room communication. J Surg Res. 2019;235:105–112.

Thank you for the suggestions. We have the references and have included those where relevant in the discussion.

---

## [Decision Letter · Decision Letter 1]

17 Oct 2023

PONE-D-23-13370R1Eyes on Incivility in Surgical Teams: Teamwork, Well-Being, and an InterventionPLOS ONE

Dear Dr. Ostroff,

Thank you for submitting your manuscript to PLOS ONE. After careful consideration, we feel that it has merit but does not fully meet PLOS ONE’s publication criteria as it currently stands. Therefore, we invite you to submit a revised version of the manuscript that addresses the points raised during the review process.

We look forward to receiving your revised manuscript.

Kind regards,

Jonas Preposi Cruz

Academic Editor

PLOS ONE

Journal Requirements:

**Additional Editor Comments:**

Thank you for submitting your revised version to the journal. You have done a fantastic job revising your manuscript based on the reviewers' comments. One reviewer has additional comments, which I invite you to address. I look forward to receiving your revised manuscript for further consideration.

Reviewers' comments:

Reviewer's Responses to Questions

**Comments to the Author**

1. If the authors have adequately addressed your comments raised in a previous round of review and you feel that this manuscript is now acceptable for publication, you may indicate that here to bypass the “Comments to the Author” section, enter your conflict of interest statement in the “Confidential to Editor” section, and submit your "Accept" recommendation.

Reviewer #1: All comments have been addressed

Reviewer #2: All comments have been addressed

2. Is the manuscript technically sound, and do the data support the conclusions?

Reviewer #1: Yes

Reviewer #2: Yes

3. Has the statistical analysis been performed appropriately and rigorously? 

Reviewer #1: Yes

Reviewer #2: Yes

4. Have the authors made all data underlying the findings in their manuscript fully available?

Reviewer #1: Yes

Reviewer #2: Yes

5. Is the manuscript presented in an intelligible fashion and written in standard English?

Reviewer #1: Yes

Reviewer #2: Yes

6. Review Comments to the Author

Reviewer #1: Thanks for the opportunity to review this manuscript. The authors have done a great job at addressing reviewer comments. I just have several minor comments for the authors that would benefit the manuscript.

Introduction

The following sentence in paragraph 2 “To add to the growing body of literature on incivility in healthcare, we investigate relationships between individuals’ experiences and observations of incivility, team dynamics, and ultimately well-being using mediated analytical models.” would be better located in the final paragraph of the introduction.

The introduction could be strengthened with mention of the culture change programs in the US and Australia (Promoting Professional Accountability Program - Vanderbilt University Medical Centre, and in Australia - Ethos program). Although mentioned in the Discussion, it could be made clearer that these programs do aim to reduce low level / low intensity behaviours such as incivility in order to reduce the likelihood of patterns of unprofessional behaviour and the gradation to more serious behaviours such as physical or sexual assault. A key to these programs is the informal feedback process in which trained staff provide feedback to those reported for unprofessional behaviour to increase awareness and provide an opportunity for change. A majority of behaviours reported in these studies are incivility behaviours, and there is evidence to suggest these programs decrease incivility. It is worth mentioning this in the introduction.

Materials and Methods

Please include the survey items as an appendix. I think this would be beneficial for the reader and those who may want to replicate the study.

“Signs were of 2 sizes (210 x 297cm and 297 x 420cm).” Do you mean mm rather than cms? 420cm is 4 meters.

Results

Please provide a citation given that your approach to examining the open ended question was thematic analysis.

Reviewer #2: Comments on the revision

C1: It would improve the paper to be more specific about these items or validated measurement tools for measuring team dynamics, and maybe also be clearer about how the three items for Job attitudes are emphasized and how they are formulated.

The authors have addressed this comment by adding details and sample items on page 7 – great.

C2: Description on the thematical analysis of the open-ended questions is needed.

The authors have included further details about the thematic analysis, beginning pg. 11 – great.

C3: Results: The authors may consider writing the same names of in the tables.

The authors have ensured consistent terms have been used throughout – great.

C4: The authors may consider writing the same names on the items. Team Dynamics in the vertical column and Team in the horizontal row. Job Attitudes in the vertical column and Attitudes in the horizontal row. I assume it is the same item. Perhaps the full name is hidden in the Table?

The authors have expanded the labels in the Table 2 to enhance clarity - great.

C5: The authors might consider adding some comments on how the identification was conducted.

The authors have stated that they have followed standard procedures for thematic analysis and have included additional details on pg. 11. – great.

C5: The authors may consider including a comment on the fact that the percentage response from the group of Admit, Day and Recovery nurses is considerably greater Time 2 (20 out of 45) than Time 1 (26 out of 74). How does it affect the results in the light of the small difference between the Pre and Post measurements in this group (Fig 3)? And maybe a comment on the significantly reduced response from the group of Surgeons, Anesthetists and Assistants Time 2.

The authors have addressed some of these limitations in the discussion. – great.

C6: The authors might consider adding some comments on how the open-ended

responses could be used in order to improve the communication in the surgical teams.

The authors have noted in the discussion that the open-ended comments highlight that incivility likely stems from a multitude of individual and broader system factors. We then link this to a discussion of wider system interventions – great.

7. PLOS authors have the option to publish the peer review history of their article (what does this mean?). If published, this will include your full peer review and any attached files.

Reviewer #1: No

Reviewer #2: No

---

## [Author Response · Author response to Decision Letter 1]

21 Oct 2023

Comments to Reviewers

Reviewer #1: Thanks for the opportunity to review this manuscript. The authors have done a great job at addressing reviewer comments. I just have several minor comments for the authors that would benefit the manuscript.

Introduction

The following sentence in paragraph 2 “To add to the growing body of literature on incivility in healthcare, we investigate relationships between individuals’ experiences and observations of incivility, team dynamics, and ultimately well-being using mediated analytical models.” would be better located in the final paragraph of the introduction.

Thank you for your suggestion. As suggested, we have moved this sentence to the final paragraph. 

The introduction could be strengthened with mention of the culture change programs in the US and Australia (Promoting Professional Accountability Program - Vanderbilt University Medical Centre, and in Australia - Ethos program). Although mentioned in the Discussion, it could be made clearer that these programs do aim to reduce low level / low intensity behaviours such as incivility in order to reduce the likelihood of patterns of unprofessional behaviour and the gradation to more serious behaviours such as physical or sexual assault. A key to these programs is the informal feedback process in which trained staff provide feedback to those reported for unprofessional behaviour to increase awareness and provide an opportunity for change. A majority of behaviours reported in these studies are incivility behaviours, and there is evidence to suggest these programs decrease incivility. It is worth mentioning this in the introduction.

We appreciate your comment and have added some further information on these programs to the introduction, highlighting that these interventions have shown a positive impact on the reduction of disruptive behaviors in the healthcare space. A limitation of these programs are the additional costs and resources needed for these systems to be successfully implemented, justifying the need for potential low-costs options (such as the eye signage) to be explored. 

Materials and Methods

Please include the survey items as an appendix. I think this would be beneficial for the reader and those who may want to replicate the study.

We did not initially include the survey items as an appendix as our items are all derived from previously used measures. However, in response we have now made the items publicly available at https://doi.org/10.17605/OSF.IO/67HNV

and have indicated this in the text.

“Signs were of 2 sizes (210 x 297cm and 297 x 420cm).” Do you mean mm rather than cms? 420cm is 4 meters.

Thank you for noticing this oversight on our behalf. We have changed this to mm. 

Results

Please provide a citation given that your approach to examining the open ended question was thematic analysis.

A citation has now been added: Braun V, Clarke V. Using thematic analysis in psychology. Qual Res Psychol. 2006 Jan 1;3(2):77-101.

---

## [Decision Letter · Decision Letter 2]

20 Nov 2023

Eyes on Incivility in Surgical Teams: Teamwork, Well-Being, and an Intervention

PONE-D-23-13370R2

Dear Dr. Ostroff,

We’re pleased to inform you that your manuscript has been judged scientifically suitable for publication and will be formally accepted for publication once it meets all outstanding technical requirements.

Kind regards,

Jonas Preposi Cruz

Academic Editor

PLOS ONE

Additional Editor Comments (optional):

Reviewers' comments:

Reviewer's Responses to Questions

**Comments to the Author**

1. If the authors have adequately addressed your comments raised in a previous round of review and you feel that this manuscript is now acceptable for publication, you may indicate that here to bypass the “Comments to the Author” section, enter your conflict of interest statement in the “Confidential to Editor” section, and submit your "Accept" recommendation.

Reviewer #1: All comments have been addressed

2. Is the manuscript technically sound, and do the data support the conclusions?

Reviewer #1: Yes

3. Has the statistical analysis been performed appropriately and rigorously? 

Reviewer #1: Yes

4. Have the authors made all data underlying the findings in their manuscript fully available?

Reviewer #1: Yes

5. Is the manuscript presented in an intelligible fashion and written in standard English?

Reviewer #1: Yes

6. Review Comments to the Author

Reviewer #1: Thanks for the opportunity to review this manuscript. The authors have addressed the reviewer comments. However, there are two typographical errors:

1. In the sentence that has been added "Initiatives such as the Promoting Professional Accountability Program [27] and Ethos program [28] have helped reduce descriptive behaviors." please change 'descriptive' to 'disruptive'.

2. In the sentence "While these programs can help reduce incivility, they are costly as successful implementation requires trained personnel to implement the surveillance system and for staff to be trained in its use [27]." please change 'surveillance' to 'reporting'.

7. PLOS authors have the option to publish the peer review history of their article (what does this mean?). If published, this will include your full peer review and any attached files.

Reviewer #1: No

---

## [Editor Report · Acceptance letter]

22 Nov 2023

PONE-D-23-13370R2 

Eyes on Incivility in Surgical Teams: Teamwork, Well-Being, and an Intervention 

Dear Dr. Ostroff:

I'm pleased to inform you that your manuscript has been deemed suitable for publication in PLOS ONE. Congratulations! Your manuscript is now with our production department. 

Kind regards, 

on behalf of

Dr. Jonas Preposi Cruz 

Academic Editor

PLOS ONE